# Wild-Type AmpC Beta-Lactamase-Producing *Enterobacterales* Are a Risk Factor for Empirical Treatment Failure in Patients with Bloodstream Infection

**DOI:** 10.3390/diseases12030052

**Published:** 2024-03-02

**Authors:** Matteo Vassallo, Roxane Fabre, Laurene Lotte, Sabrina Manni, Christian Pradier

**Affiliations:** 1Department of Internal Medicine/Infectious Diseases, Cannes General Hospital, 06400 Cannes, France; s.manni@ch-cannes.fr; 2Unité de Recherche Clinique Cote d’Azur (UR2CA), URRIS, Centre Hospitalier Universitaire, Pasteur 2, 06000 Nice, France; 3Public Health Department, Archet Hospital, Nice University, 06202 Nice, France; fabre.r@chu-nice.fr (R.F.); pradier.c@chu-nice.fr (C.P.); 4Pain Department and FHU InovPain, Nice University Hospital, Cote Azur University, 06000 Nice, France; 5Multipurpose Laboratory, Bacteriology and Virology Unit, Cannes General Hospital, 06400 Cannes, France; la.lotte@ch-cannes.fr

**Keywords:** AmpC-producing *Enterobacterales*, bloodstream infection, third-generation cephalosporins, treatment failure

## Abstract

**Introduction:** Beta-lactamases are frequently prescribed for Gram-negative bloodstream infections (BSIs). However, chromosomally encoded AmpC-producing *Enterobacterales* (AE) could overproduce beta-lactamases when exposed to third-generation cephalosporins (3GCs), with a risk of clinical failure. There are few available in vivo data on the subject. Our goal was to assess the potential role of AE as a predictive factor for clinical failure in patients with BSIs. **Materials and Methods:** We retrospectively analyzed patients admitted to Cannes hospital between 2021 and 2022 for BSIs due to *Enterobacterales*. Patient demographics, comorbidities, and main clinical and laboratory parameters during hospitalization were collected. The risk factors for clinical instability after 48 h or death, as well as for ineffective initial empirical therapy, were assessed using univariate and multivariate analyses. **Results:** From January 2021 to December 2022, 101 subjects were included (mean age 79 years, 60% men, 97% with comorbidities, 17% with healthcare-associated infection, 13% with septic shock, 82% with qPitt severity score < 2, 58% with urinary tract infection, and 18% with AE). Septic shock [adjusted odds ratio (OR_adj_) = 5.30, 95% confidence interval (CI): 1.47–22.19, *p* = 0.014] and ineffective initial empirical therapy [OR_adj_ 5.54, 95% CI: 1.95–17.01, *p* = 0.002] were independent predictive factors for clinical instability or death. Extended-spectrum beta-lactamases [OR_adj_ 9.40, 95% CI: 1.70–62.14, *p* = 0.012], AE group [OR_adj_ 5.89, 95% CI: 1.70–21.40, *p* = 0.006], and clinical instability or death [OR_adj_ 4.71, 95% CI: 1.44–17.08, *p* = 0.012] were independently associated with ineffective empirical therapy. **Conclusions**: Infection with AE was associated with treatment failure. Empirical therapy may result in failure if restricted to 3GC.

## 1. Introduction

The burden of Gram-negative bacteria resistant to third-generation cephalosporins has been highlighted by the World Health Organization [1] and can be potentially life-threatening, especially in the case of bloodstream infections (BSIs), defined by the presence of viable microorganisms in the bloodstream eliciting an inflammatory response characterized by an alteration in clinical, laboratory and hemodynamic parameters [2]. The resistance of *Enterobacterales* mainly results from the plasmid-mediated acquisition of beta-lactamase enzymes or the deregulation of natural genetically encoded enzymes [3].

Chromosomally encoded AmpC-producing beta-lactamases (AE) include *Enterobacterales* species such as *Enterobacter* spp., *Serratia marcescens*, *Citrobacter freundii*, *Providencia* spp., and *Morganella morganii* (ESCPM), which have a natural resistance to aminopenicillins and first-generation cephalosporins [3,4]. When exposed to third-generation cephalosporins (3GCs) such as ceftriaxone and cefotaxime, these can select AmpC-overproducing strains, resulting in 3GC inactivation and treatment failure [5,6,7]. 

ESCPM microorganisms are implicated in a broad range of infections involving urinary and gastrointestinal tracts, lungs, skin, and soft tissues. Although several in vitro studies describe the risk of selecting AmpC-overproducing ESCPM strains when these are exposed to 3GCs, few in vivo data are available regarding the outcome of BSIs in the case of empirical treatment with 3GCs alone [8,9]. 

The aim of this study was to analyze the possible role of ESCPM organisms as a risk factor for clinical failure in patients with BSIs.

## 2. Materials and Methods

### 2.1. Study Design and Participants

We conducted an observational, retrospective cohort study of patients admitted to any clinical department in Cannes General Hospital in 2021 and 2022 for BSIs due to *Enterobacterales*. Data from each patient group were extracted from the hospital’s electronic database. Patient demographics, underlying comorbidities, duration of symptoms, clinical signs upon admission and during hospitalization, laboratory findings during hospital stay, and clinical outcomes were collected from their medical records. In order to assess sepsis severity, the qPitt score was calculated for each patient at the time of the first positive blood culture. Briefly, it consists of five binary parameters, one point being assigned to each of the following: temperature < 36°, systolic blood pressure ≤ 90 mmHg or vasopressor administration, respiratory rate ≥ 25/min or need of mechanical ventilation, altered mental status, and cardiac arrest. Sepsis is generally considered severe for a score of 2 points or above [10].

According to the *Enterobacterales* species responsible for the BSI, patients were divided into those infected and those not infected with ESCPM organisms, as defined elsewhere [4]. As the main goal of the study was to evaluate the potential correlation between the ESCPM group and outcome, patients were excluded in the case of polymicrobial BSI. Subjects were also excluded if the clinical outcome was not available, for example, in the case of admission to the emergency department and transfer to another hospital with no available clinical file.

The study was submitted to the Health Data Hub (https://www.health-data-hub.fr/depot, accessed on 27 November 2023), and all patients received written information regarding the study and gave their consent to participate.

### 2.2. Definitions and Statistical Analysis

After describing the study population’s main characteristics, we first searched whether the initial empirical antibiotic therapy was associated with a favorable outcome. Patients were therefore divided into those presenting with clinical instability after 48 h of BSI diagnosis or dying in-hospital, and those clinically stable at 48 h.

Clinical instability after 48 h of BSI diagnosis was defined by either clinical (i.e., persistence of fever or other clinical signs of infection) or microbiological (i.e., persistence of the same bacterial species on blood cultures) criteria. The empirical antibiotic treatment was defined as effective if the *Enterobacterales* species identified on blood cultures was susceptible to at least one of the antimicrobial agents prescribed according to bacterial susceptibility testing, realized using the disk diffusion method (i2a^®^ disks, Montpellier, France) and interpreted in accordance with the European Committee on Antimicrobial Sensitivity Testing (EUCAST) guidelines. In the case of ESCPM microorganisms, 3GCs were considered ineffective. 

Comparisons were performed using χ-square or Fisher’s exact tests and Student’s *t*-tests or the Wilcoxon–Mann–Whitney test. Independent risk factors associated with clinical outcomes were identified by logistic regression.

In case ofa significant association between ineffective therapy and a worse outcome, predictive factors for ineffective initial therapy were measured via univariate and multivariate analyses using the tests described above. 

Kaplan–Meier curves were performed if parameters were significantly associated with mortality. A Cox regression analysis was used to obtain the hazard ratio. Analyses were performed using the R-4.3.0 software.

## 3. Results

### 3.1. Population Characteristics

From January 2021 to December 2022, 184 cases of BSIs due to *Enterobacterales* were collected. After excluding polymicrobial infections and cases with unavailable clinical data, 101 subjects were included in the study (mean age 79 years, 60% men, 97% with comorbid conditions including 30% with active cancer, 17% with healthcare-associated infection, 94% admitted in a medical department at the time of BSI, Table 1). Among the other comorbid conditions, the most frequent were chronic cardiopathies, diabetes, and neurocognitive disorders.

The main suspected portals of entry for BSIs were urinary and gastrointestinal infections. Septic shock was diagnosed in 13% of subjects, while the majority of individuals had a low qPitt severity score at the time of their first positive blood culture (Table 1).

Among the isolated microorganisms, ESCPM microorganisms were identified in 18% of BSIs (nine *Enterobacter* sp., five *Serratia* sp., two *Morganella* sp., one *Providencia* sp., and one *Pantoea* sp.). The majority of BSIs were due to *Escherichia coli* (41%) and *Klebsiella* sp. (30%), while extended-spectrum beta-lactamase (ESBL)-producing organisms were identified in 8% of the cases.

For 59% of the patients, the empirical initial antibiotic therapy consisted of a single drug, in most cases a 3GC. Among patients receiving combination therapy, the main prescriptions consisted of 3GCs and metronidazole, followed by 3GCs and aminoglycosides. 

### 3.2. Risk Factors for a Worse Clinical Outcome

Univariate analysis showed that clinical instability after 48 h from BSI diagnosis or in-hospital death was significantly associated with a shorter duration of antibiotic therapy, septic shock, ESCPM group, ineffective initial empirical therapy, and treatment with a single antibiotic agent (Table 2). As *Enterobacter* sp. are considered among the most AmpC-overproducing strains [11], we compared *Enterobacter* sp. with the other ESCPM group species, but we did not find any difference in clinical outcomes. 

In the multivariate analysis, the independent risk factors for clinical instability or death were septic shock [adjusted odds ratio (OR_adj_) = 5.30, 95% confidence interval (CI): 1.47–22.19, *p* = 0.014] and ineffective initial empirical therapy [OR_adj_ 5.54, 95% CI: 1.95–17.01, *p* = 0.002] (Table 2).

### 3.3. Risk Factors for Ineffective Empirical Therapy

Univariate analysis revealed that ineffective initial empirical antibiotic therapy was associated with infection due to ESBL, the ESCPM group, clinical instability after 48 h or death, and the administration of a single antibiotic agent (Table 3).

Multivariate analysis revealed that ESBL [OR_adj_ 9.4, 95% CI: 1.7–62.14, *p* = 0.012], the ESCPM group [OR_adj_ 5.89, 95% CI: 1.7–21.40, *p* = 0.006], and clinical instability after 48 h or death [OR_adj_ 4.71, 95% CI: 1.44–17.08, *p* = 0.012] were independently associated with ineffective empirical therapy (Table 3).

Table 4 provides characteristics in details of patients with ESCPM and ESBL BSIs (Table 4).

### 3.4. Analysis of Deaths

Among the 101 patients included, 24 in-hospital deaths occurred. For 20 subjects, according to our in-depth file review, causes of death were very likely due to BSIs, while for the other 4 cases, the fatal outcome was probably linked to other causes, such as COVID-19 infection or cancer.

Kaplan–Meier curves showed a significant difference in survival according to the efficacy of initial antibiotic therapy, considering either crude mortality or BSI-related deaths (Figure 1).

## 4. Discussion

In a retrospective cohort of patients admitted for BSIs due to *Enterobacterales*, we found high rates of unfavorable outcomes. Infection due to ESCPM organisms was an independent risk factor for ineffective initial antibiotic therapy.

The patients included in this study were representative of the rising rates of elderly and severely comorbid individuals presenting at emergency departments in most countries. Indeed, the British Geriatric Society reported that patients over 65 years constituted the majority of hospital admissions, bed days, and emergency readmissions, with high risks of mortality [12,13]. The increasing number of elderly populations worldwide [14,15] who are more likely to have complex presentations, multiple comorbidities, and multi-medication exposure represents a major challenge for clinical care. 

Although the majority of BSIs were not severe at the time of diagnosis, we found high rates of unfavorable outcomes. These results are in line with previous studies showing that older patients have increased mortality rates compared to their younger counterparts [16]. Indeed, elderly patients are generally more prone to infection, as a consequence of aging, comorbidities, immune system dysfunction, and the use of invasive devices [17]. Moreover, classical manifestations of systemic inflammatory response to BSIs may be minimally present in older subjects. Indeed, the febrile response can be more frequently blunted than in younger subjects and may be replaced by other less evocative signs, such as weakness, confusion, falls, or loss of appetite [17]. Moreover, the prognosis for Gram-negative BSI is particularly poor due to virulence factors and antibiotic resistance [18,19,20]. However, the prevalence of ESBL was relatively low in our study, in line with current epidemiological data in France, which show significant differences among *Enterobacterales* but low levels of ESBL in *E. coli* species, the most frequently encountered organism in this study [21,22,23]. Our results are quite similar to those recently presented by Maillard et al., who showed higher rates of treatment failure to 3GC than cefepime in a retrospective analysis of patients with BSIs and pneumonia caused by AE. However, the authors did not find any difference in terms of mortality, probably because half of the patients included did not have sepsis [8].

If confirmed by larger and prospective studies, empirical antibiotic therapy for BSIs should take into account the ESCPM group and the possibility that single therapy with 3GCs could be associated with risks of failure. Therefore, we suggest that in cases of in-patients with a suspected BSI due to *Enterobacterales* or to positive blood cultures for Gram-negative bacteria, pending identification and susceptibility testing, cefepime should replace 3GCs as empirical treatment if a single compound is chosen, or alternatively, a double Gram-negative coverage with the addition of aminoglycosides could reduce risks of failure. Indeed, cefepime is a poor inducer of AmpC Beta-lactamases, and although comparisons with 3GCs of the effect on intestinal microbiota have not been specifically studied, McKinnell et al. showed lower rates of vancomycin-resistant Enterococci after cefepime than ceftriaxone use [24]. Moreover, cefepime use for BSIs due to AE is supported by recent data presented by Hermann et al., who showed its usefulness for treating BSIs if ESBL production can be excluded [25]. 

However, whether combination antimicrobial therapy is superior to monotherapy remains a controversial issue, and the main guidelines recommend limiting the use of combination therapies to critically ill patients or those with a high risk of multidrug-resistant pathogens [26,27]. Moreover, in a meta-analysis of monotherapy vs. beta-lactam-aminoglycoside combination strategy for sepsis, Paul et al. found no difference in efficacy but more frequent side effects in the case of combination therapy [28].

This study has many limitations, including its retrospective nature and the relatively small number of subjects included. Moreover, although the qPitt score previously showed good performance for predicting mortality [10], and patient files were thoroughly reviewed, we cannot exclude the risk of misinterpreting sepsis severity criteria. Furthermore, the study design did not allow microbiological confirmation that exposure to 3CGs increases the risk of developing resistant ESCMP strains, as published by Choi et al. [29].

## 5. Conclusions

In conclusion, in a population of elderly patients hospitalized for BSIs, we found that infection with ESCPM microorganisms was a predictive risk factor for treatment failure. Empirical therapy, regardless of initial clinical severity, should take into account the risk of failure in the case of 3GC monotherapy.

## Figures and Tables

**Figure 1 diseases-12-00052-f001:**
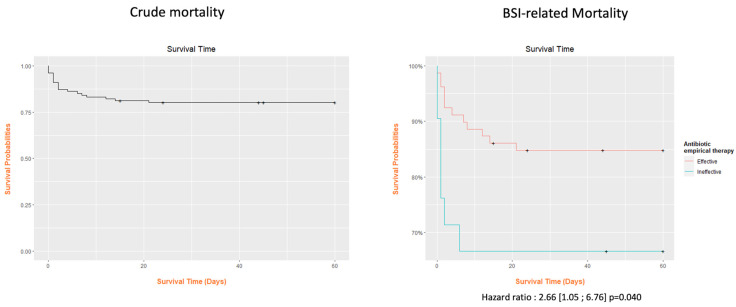
Kaplan–Meier analysis of overall mortality and bloodstream infection-related mortality according to the initial effective therapy.

**Table 1 diseases-12-00052-t001:** Patient characteristics.

	N (%) or Mean [SD]
**Number of patients**	101
**Age (years)**	78.6 [12.1]
**Male gender**	61 (67.4%)
**With active cancer**	30 (29.7%)
**Number of comorbid conditions**	
0	3 (3.0%)
1	25 (24.8%)
2	37 (36.6%)
≥3	36 (35.6%)
**Healthcare-associated infection**	17 (17.0%)
**Septic shock associated with BSI**	13 (12.9%)
**Department at the time of BSI diagnosis**	
Medical Department	95 (94.1%)
Intensive Care Unit	6 (5.9%)
**Site of infection**	
Urinary tract	59 (58.4%)
Gastrointestinal	22 (21.8%)
Pulmonary	6 (5.9%)
Vascular catheter	5 (5.0%)
Other	9 (8.9%)
**Type of *Enterobacterales***	
ESCPM group	18 (17.8%)
Other *Enterobacterales*	83 (82.2%)
**qPitt score for BSI severity**	
<2	83 (82.2%)
≥2	18 (17.8%)
**ESBL**	8 (8.2%)
**Effective empirical treatment**	
No	21 (21.0%)
Yes	79 (79.0%)
**Type of empirical antimicrobial therapy**	
Single therapy	60 (62.5%)
Combination therapy	36 (37.5%)
**Clinical stability at 48 h**	
No	32 (31.7%)
Yes	69 (68.3%)
**In-hospital death**	24 (23.8%)
**Death probably related to BSI**	
No	4 (16.7%)
Yes	20 (83.3%)
**Clinical instability at 48 h or death**	36 (35.6%)
**Recurrent BSI within 14 days**	3 (3.0%)

BSI: bloodstream infection; ESCPM group: *Enterobacter* spp., *Serratia marcescens*, *Citrobacter freundii*, *Providencia* spp., and *Morganella morganii*.

**Table 2 diseases-12-00052-t002:** (a) Risk factors for clinical instability after 48 h of therapy or in-hospital death—univariate analysis; (b) independent risk factors for clinical instability after 48 h of therapy or in-hospital death.

(a)					
	Clinically Stable at 48 h and No Death n = 65		Clinically Unstable at 48 h or Death n = 36		
	Mean	[SD]	Mean	[SD]	*p*-Value
**Age**	79.9	[11.3]	76.3	[13.2]	0.249
**Days of hospitalization**	12.8	[10.1]	12.1	[10.9]	0.524
**Delay until reassessment of empirical therapy (days)**	3.6	[1.9]	3.3	[2.7]	0.201
**Days of antibiotic therapy**	14.0	[7.1]	10.4	[9.5]	0.010
**qPitt score**	0.5	[0.77]	0.8	[0.95]	0.123
**White-cell count (cc/mmcc)**	13,622.2	[8757.7]	14,068.3	[7650.2]	0.618
**C-reactive protein (mg/L)**	122.6	[96.8]	153.6	[123.0]	0.242
**Procalcitonin (ng/mL)**	31.6	[52.3]	38.5	[59.8]	0.777
	**n**	**(%)**	**n**	**(%)**	***p*-Value**
**Gender**					0.244
Female	23	(35.4)	17	(47.2)	
Male	39	(34.2)	26	(59.1)	
**Comorbidities**					1.000
No	2	(3.1)	1	(2.8)	
Yes	63	(96.9)	35	(97.2)	
**Healthcare associated BSI**					0.240
No	51	(79.7)	32	(88.9)	
Yes	13	(20.3)	4	(11.1)	
**ICU at BSI diagnosis**					0.663
No	62	(95.4)	33	(91.7)	
Yes	3	(4.6)	3	(8.3)	
**qPitt score**					0.257
<2	56	(86.2)	27	(75.0)	
≥2	9	(13.8)	9	(25.0)	
**Septic shock**					0.011
No	61	(92.3)	27	(75.0)	
Yes	4	(6.2)	9	(25.0)	
**ESBL**					1.000
No	60	(92.3)	30	(90.9)	
Yes	5	(7.7)	3	(9.1)	
**Type of *Enterobacterales***					0.013
ESCPM group	7	(10.8)	11	(30.6)	
Other	58	(89.2)	25	(71.4)	
**Effective empirical therapy**					0.001
No	7	(10.8)	14	(40.0)	
Yes	58	(89.2)	21	(60.0)	
**Type of empirical therapy**					0.015
Single therapy	46	(70.8)	14	(45.2)	
Combination therapy	19	(29.2)	17	(54.8)	
**(b)**					
	**AdjOR**		**[95% CI]**		***p*-Value**
**Septic shock**					
No	1				
Yes	5.30		[1.47; 22.19]		0.014
**Initial empirical antibiotic therapy**					
Ineffective	5.54		[1.95; 17.01]		0.002
Effective	1				

**Table 3 diseases-12-00052-t003:** (a) Risk factors for initial ineffective empirical antibiotic therapy—univariate analysis *; (b) independent factors associated with ineffective empirical antibiotic therapy.

(a)					
	Ineffective Empirical Therapy n = 21		Effective Empirical Therapy n = 79		
	Mean	[SD]	Mean	[SD]	*p*-Value
**Age (years)**	76.9	[10.4]	79.0	[12.5]	0.275
**Days of hospitalization**	11.1	[9.7]	12.8	[10.5]	0.592
**Delay until reassessment of empirical therapy (days)**	3.1	[1.4]	3.6	[2.3]	0.827
**White-cell count (cc/mmcc)**	12,107.6	[6770.2]	14,416.6	[8599.6]	0.341
**C-reactive protein (mg/L)**	156.3	[145.5]	126.4	[95.3]	0.775
**Procalcitonin (ng/mL)**	60.1	[100.2]	31.0	[47.6]	0.975
	**n**	**(%)**	**n**	**(%)**	* **p** * **-Value**
**Gender**					0.924
Female	8	(38.1)	31	(39.2)	
Male	13	(61.9)	48	(60.8)	
**ICU when BSI**					0.603
No	19	(90.5)	75	(94.9)	
Yes	2	(9.5)	4	(5.1)	
**Septic shock**					0.464
No	17	(81.0)	70	(88.6)	
Yes	4	(6.2)	9	(25.0)	
**ESBL**					0.008
No	15	(75.0)	75	(96.2)	
Yes	6	(25.0)	3	(3.8)	
**Type of *Enterobacterales***					<0.001
ESCPM group	10	(47.6)	8	(10.1)	
Other	11	(52.4)	71	(89.9)	
**Clinically unstable at 48 hr or death**					
No	7	(33.3)	58	(73.4)	
Yes	14	(66.7)	21	(26.6)	
**Type of empirical therapy**					0.001
Single therapy	10	(47.6)	49	(62.0	
Combination therapy	6	(28.6)	30	(38.0)	
No empirical therapy	5	(23.8)	0	(0.0)	
**(b)**					
	**AdjOR**		**[95% CI]**		***p*-Value**
**ESBL**					
No	1				
Yes	9.40		[1.70; 62.14]		0.012
**Type of *Enterobacterales***					
ESCPM group	5.89		[1.70; 21.40]		0.006
Other	1				
**Clinical instability at 48 hr or death**					
No	1				
Yes	4.71		[1.44; 17.08]		0.012

* One patient was excluded as she died a few hours after the diagnosis of sepsis, and antibiotic sensitivity testing was not available. ESBL: extended-spectrum beta-lactamase; ESCPM: *Enterobacter* sp., *Serratia* sp., *Citrobacter* sp., *Providencia* sp., *Morganella* sp.

**Table 4 diseases-12-00052-t004:** (a) Characteristics of patients with ESCPM BSIs and (b) characteristics of patients with ESBL BSIs.

**(a)**												
**Patient**	**Sex**	**Age (years)**	**Number of Comorbidities**	**ESCPM Species**	**ESBL**	**Suspected Site Associated witd BSI**	**Healthcare-Associated Infection**	**qPitt Score for Severity**	**Initial Empirical Therapy according to AST**	**Clinical Stability after 48 h**	**Delay Until Reassessment of Therapy in Case of Switch (days)**	**In-Hospital Death**
1	M	65	2	*Enterobacter cloacae*	Yes	Urinary	No	0	Effective	Yes	No change	No
2	M	69	2	*Enterobacter cloacae*	No	Gastrointestinal	No	0	Ineffective	Yes	5	No
3	F	77	3	*Klebsiella aerogenes*	No	Vascular	No	0	Ineffective	No	3	No
4	M	72	2	*Enterobacter cloacae*	No	Pulmonary	Yes	2	Ineffective	Yes	No change	Yes
5	M	88	4	*Enterobacter cloacae*	Yes	Urinary	No	0	Ineffective	No	5	No
6	F	69	1	*Enterobacter cloacae*	No	Gastrointestinal	No	1	Ineffective	No	No change	Yes
7	M	89	2	*Enterobacter cloacae*	Yes	Vascular	Yes	0	Effective	Yes	2	No
8	F	70	1	*Enterobacter cloacae*	No	Gastrointestinal	No	2	Effective	Yes	No change	No
9	M	58	1	*Enterobacter cloacae*	No	Gastrointestinal	No	0	Ineffective	Yes	1	No
10	M	56	1	*Providencia stuartii*	No	Urinary	No	2	Ineffective	No	3	No
11	F	85	2	*Morganella morgnaii*	No	Urinary	No	0	Effective	Yes	No change	No
12	F	85	3	*Pantoea agglomerans*	No	Vascular	No	0	Effective	No	2	No
13	M	91	1	*Morganella morgnaii*	No	Urinary	No	1	Effective	No	1	Yes
14	M	81	3	*Serratia marcescens*	No	Gastrointestinal	No	0	Ineffective	No	3	No
15	M	69	2	*Serratia marcescens*	No	Skin and soft tissues	Yes	0	Effective	Yes	No change	No
16	M	77	4	*Serratia marcescens*	No	Gastrointestinal	Yes	0	Ineffective	No	No change	No
17	F	79	3	*Serratia marcescens*	No	NA	No	1	Ineffective	No	No change	Yes
18	M	54	2	*Serratia marcescens*	No	Vascular	No	0	Effective	No	10	Yes
**(b)**												
**Patient**	**Sex**	**Age**	**Number of Comorbidities**	***Enterobacterales* Species**		**Suspected Site Associated with BSI**	**Healthcare-Associated Infection**	**qPitt Score for Severity**	**Initial Empirical Therapy**	**Clinical Stability after 48 h**	**Delay until Reassessment of Therapy in Case of Switch (days)**	**In-Hospital Death**
1	M	65	2	*Enterobacter cloacae*		Urinary	No	0	Effective	Yes	No change	No
2	M	83	3	*Escherichia coli*		Pulmonary	No	2	Ineffective	No	2	Yes
3	M	88	4	*Enterobacter cloacae*		Urinary	No	0	Ineffective	No	5	No
4	M	89	2	*Enterobacter cloacae*		Vascular	Yes	0	Effective	Yes	2	No
5	F	63	2	*Escherichia coli*		Urinary	No	0	Ineffective	No	2	No
6	F	83	2	*Klebsiella pneumoniae*		Urinary	Yes	1	Effective	Yes	No change	No
7	F	88	3	*Escherichia coli*		Gastrointestinal	No	2	Ineffective	Yes	4	No
8	F	78	2	*Escherichia coli*		Urinary	No	0	Ineffective	Yes	5	No

ESCPM: *Enterobacter* sp., *Serratia marcescens*, *Citrobacter freundii*, *Providencia* sp., *Morganella morganii*; ESBL: extended-spectrum beta-lactamase; BSI: bloodstream infection; AST: antibiotic sensitivity testing.

## Data Availability

Data respecting patient’s anonymity are available if necessary.

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
