# Peer review of "Wild-Type AmpC Beta-Lactamase-Producing Enterobacterales Are a Risk Factor for Empirical Treatment Failure in Patients with Bloodstream Infection"

_diseases, 2024, doi:10.3390/diseases12030052_

Round 1

Reviewer 1 Report

Comments and Suggestions for Authors

Breief summary

The authors focused on the AmpC-producing Enterobacterales, ESCPM, and analyzed 101 cases of BSI and showed that antimicrobial therapy with third-generation cephalosporins for BSI caused by these bacteria tended to result in failure of eradication and worsening clinical course.

General concept comments

Table 1 shows that among 101 cases, 18 are in the ESCPM group. However, Table 3 shows 11 cases in the non-ESCPM group and 71 cases in the ESCPM group. 
It is suspected that one of the tables shows the ESCPM and non-ESCPM groups incorrectly. 

Please check carefully to see if this change causes any serious breakdown in the text.

Also, since Table 3 shows 100 total cases, one case was excluded and there should be a rational reason in the text for excluding that one case.

Specific comment

L38 The font of the reference number is wrong.

In both Table 2 and Table 3, a table of independent risk factors is included as a postscript. It would be easier to read if this section were split into Table 2a and Table 2b with a space between them.

Author Response

General concept comments

Table 1 shows that among 101 cases, 18 are in the ESCPM group. However, Table 3 shows 11 cases in the non-ESCPM group and 71 cases in the ESCPM group. 
It is suspected that one of the tables shows the ESCPM and non-ESCPM groups incorrectly. 

Please check carefully to see if this change causes any serious breakdown in the text.

Also, since Table 3 shows 100 total cases, one case was excluded and there should be a rational reason in the text for excluding that one case.

We are sorry for the confusion. In Table 3 values for ESCPM and other Enterobacterales have been inverted and now have been modified. The reason for including only 100 subjects here is that for one patient the sensitivity testing was not available, as she died few hours later the diagnosis of sepsis and it was no longer possible to perform the test when this study was designed. We added a sentence for explanations

Specific comment

L38 The font of the reference number is wrong.

The font has been modified

In both Table 2 and Table 3, a table of independent risk factors is included as a postscript. It would be easier to read if this section were split into Table 2a and Table 2b with a space between them.

Table 2 and 3 have been split in a and b

Reviewer 2 Report

Comments and Suggestions for Authors

MDPI

Diseases

Manuscript: MDPI-2853346

The authors studied wild-type AmpC B-lactamase-producing Enterobacterales are a risk factor for empirical treatment failure in patients with bloodstream infection.

The authors presented patient characteristics (Table 1), risk factors for clinical instability (Table 2), risk factors (Table 3); Kaplan-Meier curves (Fig. 1). The authors concluded and reported that infection with ESCPM group was a predictive risk for treatment failure, Empirical therapy, regardless of initial clinical severity, should take into account a risk of failure in case of 3GC monotherapy. The authors have provided novel information in Enterbacterales.

.

Author Response

Thank you for your approval

Reviewer 3 Report

Comments and Suggestions for Authors

In the present study, the authors investigated the associations between empirical treatment with  third-generation cephalosporins (3GCs) and outcomes of patients with bloodstream infections due to AmpC-producing Enterobacterales. The introduction is informative, the methods are appropriate and the statistical analysis adequate. The Conclusion and Discussion is consistent with the results.The Discussion section may be improved. The references cited are suitable.

Major comments:
There are recent studies on the treatment of BSIs due to by wild-type AmpC-producing Enterobacterales
that the authors may compare their results. These observations can be added in the Discussion section.
1. Alexis Maillard, Tristan Delory, Juliette Bernier, Antoine Villa, Khalil Chaibi, Lélia Escaut, Adrien Contejean, Beatrice Bercot, Jérôme Robert, Fatma El Alaoui, Jacques Tankovic, Hélène Poupet, Gaëlle Cuzon, Matthieu Lafaurie, Laure Surgers, Adrien Joseph, Olivier Paccoud, Jean-Michel Molina, Alexandre Bleibtreu,Effectiveness of third-generation cephalosporins or piperacillin compared with cefepime or carbapenems for severe infections caused by wild-type AmpC β-lactamase-producing Enterobacterales: A multi-centre retrospective propensity-weighted study,International Journal of Antimicrobial Agents,
Volume 62, Issue 1,2023,106809,ISSN 0924-8579,https://doi.org/10.1016/j.ijantimicag.2023.106809.
(https://www.sciencedirect.com/science/article/pii/S0924857923000882)

2. Herrmann, J., Burgener-Gasser, AV., Goldenberger, D. et al. Cefepime versus carbapenems for treatment of AmpC beta-lactamase-producing Enterobacterales bloodstream infections. Eur J Clin Microbiol Infect Dis 43, 213–221 (2024). https://doi.org/10.1007/s10096-023-04715-5.

Minor comments
-The authors use unecessary capitals for the first letter ,e.g. line 45 'Ceftriaxone and Cefotaxime'', line 121: Metronidazole, ....Aminoglycosides.

Comments on the Quality of English Language

Minor editing of English language required

Author Response

Major comments:
There are recent studies on the treatment of BSIs due to by wild-type AmpC-producing Enterobacterales
that the authors may compare their results. These observations can be added in the Discussion section.
1. Alexis Maillard, Tristan Delory, Juliette Bernier, Antoine Villa, Khalil Chaibi, Lélia Escaut, Adrien Contejean, Beatrice Bercot, Jérôme Robert, Fatma El Alaoui, Jacques Tankovic, Hélène Poupet, Gaëlle Cuzon, Matthieu Lafaurie, Laure Surgers, Adrien Joseph, Olivier Paccoud, Jean-Michel Molina, Alexandre Bleibtreu,Effectiveness of third-generation cephalosporins or piperacillin compared with cefepime or carbapenems for severe infections caused by wild-type AmpC β-lactamase-producing Enterobacterales: A multi-centre retrospective propensity-weighted study,International Journal of Antimicrobial Agents,
Volume 62, Issue 1,2023,106809,ISSN 0924-8579,https://doi.org/10.1016/j.ijantimicag.2023.106809.
(https://www.sciencedirect.com/science/article/pii/S0924857923000882)

2. Herrmann, J., Burgener-Gasser, AV., Goldenberger, D. et al. Cefepime versus carbapenems for treatment of AmpC beta-lactamase-producing Enterobacterales bloodstream infections. Eur J Clin Microbiol Infect Dis 43, 213–221 (2024). https://doi.org/10.1007/s10096-023-04715-5.

We added the references in the discussion session

Minor comments
-The authors use unecessary capitals for the first letter ,e.g. line 45 'Ceftriaxone and Cefotaxime'', line 121: Metronidazole, ....Aminoglycosides.

Text has been modified

Reviewer 4 Report

Comments and Suggestions for Authors

This manuscript describes a retrospective study to find risk factor for BSI. They found ESCPM infection was associated with treatment failure.

First of all, there are serious misuse of the term Enterobacterales. Enterobacterales is an order of Gram-negative bacteria with the class Gammaproteobacteria. About 70 genera are included in this order. Seven genera of order Enterobacterales are listed as ESCPM that produce inducible beta-lactamase activity that is chromosomally mediated. However, in this manuscript, Enterobacterales is used as equivalent with ESCPM (lines 20, 42, 69, 88, 105, 160, 186).AE is used in line 18, 26, 29, 31.

Due to the limited number of subjects, it may not be possible to get significant result from detailed analysis of ESCPM cases. However, it is strongly recommended to make a table for each 18 ESCPM cases that includes antibiotics prescribed, AST data (if availableA), Delay until reassessment of empirical therapy, HAI, etc. Similar table for ESBL producing organisms will be valuable.

Figure 1 should be revised. The graph is not legible and there is no figure legend included.  

Minor points.

Line 28; Extended-spectrum (unbold)

Line 49; Ampc > AmpC

Line 127; Ampc > AmpC

Author Response

First of all, there are serious misuse of the term Enterobacterales. Enterobacterales is an order of Gram-negative bacteria with the class Gammaproteobacteria. About 70 genera are included in this order. Seven genera of order Enterobacterales are listed as ESCPM that produce inducible beta-lactamase activity that is chromosomally mediated. However, in this manuscript, Enterobacterales is used as equivalent with ESCPM (lines 20, 42, 69, 88, 105, 160, 186).AE is used in line 18, 26, 29, 31.

We are sorry for the confusion. We agree with the reviewer that Enterobacterales is not equivalent with species among the ESCPM group. We reformulated the sentence from line 42 to 45 in order to precise that only species listed here are among AE.

Due to the limited number of subjects, it may not be possible to get significant result from detailed analysis of ESCPM cases. However, it is strongly recommended to make a table for each 18 ESCPM cases that includes antibiotics prescribed, AST data (if availableA), Delay until reassessment of empirical therapy, HAI, etc. Similar table for ESBL producing organisms will be valuable.

We added Table 4a and 4b describing main characteristics of patients with ESCPM and ESBL infections, respectively

Figure 1 should be revised. The graph is not legible and there is no figure legend included. 

Figure 1 has been modified as requested.

Minor points.

Line 28; Extended-spectrum (unbold)

Line 49; Ampc > AmpC

Line 127; Ampc > AmpC

Text has been modified

Reviewer 5 Report

Comments and Suggestions for Authors

The manuscript of Vassallo and co-authors (Wild-Type AmpC B-Lactamase-Producing Enterobacterales Are a Risk Factor for Empirical Treatment Failure in Patients with Bloodstream Infection) deals with the risk of AmpC producing Enterobacterales as a risk factor for elderly patients. .

Here, a literature based study including the cases of 101 patients of the years 2021 to 2022.

This is based on the finding of "blood stream infections" with bacteria that produce AmpC. In this manuscript, however, there is no information about what these assumptions refer to. However, it is advisable to include this. Furthermore, it is stated that poly-microbial infections were excluded. Again, there is no information on this. Figure 1 shows the mortality rate of patients in relation to those with BSI. However, this figure lacks a legend, making it difficult to categorize it correctly without information. Detailed information should be added. In the second last paragraph, the authors summarize the overall findings of the study. This naturally raises the question of why these are not corrected beforehand.

Author Response

The manuscript of Vassallo and co-authors (Wild-Type AmpC B-Lactamase-Producing Enterobacterales Are a Risk Factor for Empirical Treatment Failure in Patients with Bloodstream Infection) deals with the risk of AmpC producing Enterobacterales as a risk factor for elderly patients. .

Here, a literature based study including the cases of 101 patients of the years 2021 to 2022.

This is based on the finding of "blood stream infections" with bacteria that produce AmpC. In this manuscript, however, there is no information about what these assumptions refer to. However, it is advisable to include this

We added the definition of BSI in the introduction session and added the appropriate reference

Furthermore, it is stated that poly-microbial infections were excluded. Again, there is no information on this.

We did not include polymicrobial infections, as it would not have been possible to analyse the correlation between AE group, empirical antibiotic therapy and clinical response. Moreover, as polymicrobial infections due to Enterobacterales generally have higher risks to be associated with surgical complications, such as gastrointestinal perforation or cholangitis, we think that including only monomicrobial infections could reduce risk of bias for interpreting final results.   

Figure 1 shows the mortality rate of patients in relation to those with BSI. However, this figure lacks a legend, making it difficult to categorize it correctly without information. Detailed information should be added. In the second last paragraph, the authors summarize the overall findings of the study. This naturally raises the question of why these are not corrected beforehand.

Figure 1 has been modified

Round 2

Reviewer 1 Report

Comments and Suggestions for Authors

>We are sorry for the confusion. In Table 3 values for ESCPM and other Enterobacterales have been inverted and now have been modified. The reason for including only 100 subjects here is that for one patient the sensitivity testing was not available, as she died few hours later the diagnosis of sepsis and it was no longer possible to perform the test when this study was designed. We added a sentence for explanations

I could not find the additional sentence. Please highlight the corrected statement in yellow.

Author Response

Dear Reviewer,

The sentence about the reason for excluding 1 patient has now been highlighted in yellow, in Table 3

Sincerely

Matteo Vassallo

Reviewer 4 Report

Comments and Suggestions for Authors

Accept in present form

Author Response

Thank you

Sincerely

Matteo Vassallo

Reviewer 5 Report

Comments and Suggestions for Authors

The modified version is fine.

Author Response

Thank you

Sincerely

Matteo Vassallo